# USING LARGE LANGUAGE MODELS FOR BUILDING FUZZY COGNITIVE MAPS

**Alina Petukhova**
COPELABS
Lusofona University
Lisbon, Portugal
`alina.petukhova@ulusofona.pt`

**Anna Kovalenko**
Department of Data Analysis
and Artificial Intelligence
Kuban State University
Krasnodar, Russia
`kovalenko.a@fpm.kubsu.ru`

**Natalia Chubyr**
Department of Applied Mathematics
Kuban State University
Krasnodar, Russia
`dissovet.11@kubsu.ru`

**Veronika Rastorgueva**
Department of Applied Mathematics
Kuban State University
Krasnodar, Russia
`veronikarastorgyeva2005@gmail.com`

## ABSTRACT

Fuzzy Cognitive Maps (FCMs) are widely used in the modelling of complex systems, reflecting the causal relationships between key concepts. The introduction of large language models (LLMs) presents new opportunities for constructing FCMs by using large text corpora and the deep learning capabilities built into these models. This study introduces a reproducible, multi-domain framework for FCM inference using large language models. Specifically, we investigate OpenAI's GPT-4 alongside other state-of-the-art LLMs to automate the construction of FCMs. Our method combines Chain-of-Thought reasoning, self-consistency sampling, and majority-vote aggregation to improve robustness. The framework is tested across three diverse domains—retail system, urban development, and brain tumor characterization. The evaluation includes a structured error taxonomy (missing, incorrect, and overestimated links), providing transparency and interpretability. Results show that LLMs can extract meaningful causal structures but also highlight limitations requiring expert validation.

## 1 INTRODUCTION

Traditionally, FCMs have been created based on expert knowledge, data, or hybrid methods Papageorgiou (Papageorgiou, 2014). However, the growing capabilities of LLMs offer a new paradigm—leveraging text corpora at scale and advanced reasoning capabilities to generate causal structures.

LLMs such as GPT OpenAI et al. (2023), LLaMA (Touvron & et al., 2023), and Falcon (Almazrouei et al., 2023), have demonstrated exceptional abilities in extracting semantic relationships and synthesising domain knowledge. These characteristics make them particularly well-suited for automated FCM construction, reducing reliance on manual input while maintaining interpretability.

One of the key advantages of LLMs is their ability to solve complex reasoning tasks, including causal analysis, analogies, and commonsense understanding (Talmor & et al., 2019; Kojima & et al., 2023). OpenAI's GPT models demonstrate high efficiency in structured reasoning and knowledge extraction. Recent studies examine how these models can be fine-tuned or how prompts can be adjusted for tasks requiring intensive reasoning, such as generating scientific hypotheses (Thoppilan & et al., 2022), decision support systems Lawless et al. (2024), and cognitive modelling Binz & Schulz (2023). In particular, methods like chain-of-thought (CoT) reasoning (Wei & et al., 2022) and self-consistency reasoning (Wang & et al., 2023) have significantly improved the performance of LLMs in structured decision-making, making them highly relevant for generating FCMs. These

approaches allow the models to gradually form causal relationships, which aligns well with the principles of FCM construction.

In addition to earlier foundational LLMs, a wave of more recent open-source models has been developed, each tailored to specific tasks such as reasoning, instruction following, code generation, and multilingual processing. These models bring enhanced capabilities that support more accurate and context-aware construction of FCMs.

Recent advances have introduced a range of high-performing models that further expand the capabilities in causal reasoning and semantic integration. The Phi-4:14B (Abdin et al., 2024), developed by Microsoft, is notable for its compact size and strong performance on reasoning tasks, particularly when enhanced with self-consistency sampling. The Qwen2.5:14B Hui et al. (2024), from Alibaba, builds on instruction tuning and multilingual capabilities, offering robust generalisation across domains. The LLaMA3:8B Grattafiori et al. (2024), released by Meta, represents a lightweight version of the LLaMA-3 series, achieving competitive performance through optimised training on diverse datasets. The Mistral-small Jiang et al. (2023), developed by Mistral AI, is engineered for efficiency and speed while maintaining strong reasoning performance. Its architecture supports sliding window attention and grouped-query attention, which enables scalable deployment in causal discovery tasks. The Gemma3:12B Team et al. (2024), released by Google DeepMind, is fine-tuned for balanced performance across reasoning, code generation, and factual consistency, with open-weight availability supporting transparent benchmarking. The deepseek-r1 DeepSeek-AI et al. (2025), from DeepSeek, integrates enhanced long-context reasoning and instruction alignment, making it particularly effective for capturing hierarchical or delayed causal dependencies in FCMs.

The gpt-4.5 OpenAI (2025a), OpenAI's model, trained using unsupervised learning, to improve pattern recognition, connection drawing, and insight generation without reasoning. This method was combined with supervised fine-tuning and reinforcement learning from human feedback. The o3 OpenAI (2025b) is OpenAI's most powerful reasoning model trained as reflective generative transformer designed specifically for complex tasks and in-depth analysis across general NLP tasks. LLMs ability to analyse semantic similarity and uncover hidden patterns in large datasets makes them a powerful tool for constructing and optimising FCMs.

Using LLMs will enhance the scalability, efficiency, and adaptability of FCMs, opening up new possibilities for decision support and scenario analysis across various domains.

LLMs are starting to be used to generate or refine FCMs across a range of domains. Maratea et al. (2022) investigates the automatic construction of FCMs from enriched textual sources, including Twitter conversations and online discussions. In the study, the authors examined how enriched corpora can improve the identification of concepts and the extraction of relationships within the context of social media analysis. This approach demonstrated that LLMs are capable of capturing public discourse and structuring it into interpretable causal maps. However, they highlighted significant challenges, particularly in dealing with the brevity, ambiguity, and noise typical of social media content—factors which often result in incomplete or spurious causal links.

Another work Berijanian et al. (2024) proposed graph-based similarity metrics alongside Elo-style ranking systems to evaluate FCMs generated by LLMs. While fine-tuning improved alignment with human judgments, the authors noted the inherent difficulty in capturing nuanced causal semantics. They also stressed the need for 'soft' similarity measures that can account for partial or uncertain matches. A further challenge identified was the difficulty of validating automatically generated maps in the absence of a clearly defined ground truth.

Schuerkamp et al. (2025) used LLMs to reconcile conflicting FCMs constructed by multiple experts. The approach successfully resolved up to 85% of the identified dissonance, suggesting that LLMs can facilitate the integration of diverse perspectives. Nevertheless, the study revealed an important limitation: LLMs may prioritise consensus at the expense of preserving meaningful minority viewpoints or context-specific subtleties, potentially diminishing the explanatory richness of the resulting maps.

These studies highlight the viability of using LLMs for FCM generation, while also drawing attention to several persistent challenges. These include the hallucination of irrelevant concepts or causal links, overfitting to linguistic patterns present in the training data, and a limited ability to account for domain-specific constraints or contextual subtleties. In addition, validating FCMs produced by

LLMs remains a non-trivial task, often relying on expert evaluation or proxy metrics that may not accurately reflect real-world reliability.

Building on this prior work, the present article continues this line of research by applying newer LLMs to previously unexplored domains and introducing a novel algorithm for the automatic inference of FCM connection weights. We position our work as a reproducible multi-domain FCM inference framework, where we explicitly combine Chain-of-Thought reasoning, self-consistency sampling, and voting aggregation into a unified pipeline. Unlike prior work, which often relied on fine-tuned or domain-specific models, we demonstrate how general-purpose LLMs can be systematically benchmarked across domains. Validation is carried out by comparing the generated maps with expert-constructed counterparts, providing a structured assessment of their accuracy and coherence.

## 2 METHODS

After the identification of key concepts by experts, the next step in the construction of FCM is to determine the causal relationships between them. In this process, experts are asked to define direct dependencies and the degree of influence between each pair of identified concepts, which is time-consuming. The use of LLMs could simplify this task by analysing text data to uncover both explicit and implicit causal relationships.

### 2.1 DATASET PREPARATION AND SELECTION

We selected three representative FCMs from distinct domains: retail system Petukhova (2022), urban development Petukhova et al. (2024), and brain tumor characterization Papageorgiou et al. (2008). These were chosen to ensure diversity in scope: (i) a business-oriented, operational decision-making domain, (ii) a socio-technical and policy-driven domain, and (iii) a sensitive biomedical domain requiring high accuracy. This diversity enables a comprehensive evaluation of LLM generalisation and adaptability across domains with varying data structures, terminology, and conceptual granularity. The resulting causal relationships were converted to adjacency matrices and compared with FCMs obtained through expert surveys to assess accuracy and completeness. This ensured comparability between model-generated outputs and expert references. The rationale for this selection was to cover heterogeneous domains with varying causal dynamics, allowing us to test generalisation and robustness of LLM-driven inference.

### 2.2 PROMPTING STRATEGY

We employed a unified prompting approach where the LLM was asked to infer causal links across all concept pairs in a single structured query. This strategy was chosen for reproducibility and efficiency, as it ensures all models receive identical instructions and produce outputs in a consistent JSON format. Importantly, we did not use pairwise prompting (separate prompts per concept pair) because it introduces prohibitive computational overhead for larger FCMs, creates inconsistencies across prompts, and causes difficulty in applying self-consistency and voting across fragmented outputs. By contrast, unified prompting ensures transparency and easier replication of the full adjacency matrix.

### 2.3 STRUCTURED REASONING TECHNIQUES

To improve precision and robustness of link discovery, LLMs were guided using Chain-of-Thought (CoT) reasoning Wei & et al. (2022), which decomposes causal inference into stepwise logical steps, increasing interpretability. To mitigate stochastic variability, we applied the self-consistency sampling method Wang & et al. (2023), generating multiple outputs for the same query. Final results were aggregated via majority-vote, where a connection was included if at least two out of three runs identified it. Where multiple weights were proposed, the median was taken. This pipeline forms the core of our reproducible FCM inference framework.

## 2.4 Implementation details

All experiments were run in Google Colab Google (2025) with an NVIDIA T4 GPU NVIDIA (2025). Model inference was performed with a temperature setting of 0 and three different random seeds (123, 345, 567). By reporting seeds and configuration parameters, we ensure reproducibility. The algorithmic pipeline is illustrated in Figure[1]. To produce the final result, a voting mechanism was applied using the following logic: if at least two out of the three results identified a connection, it was included in the final table. Where a connection was present, the median value of the results was used.

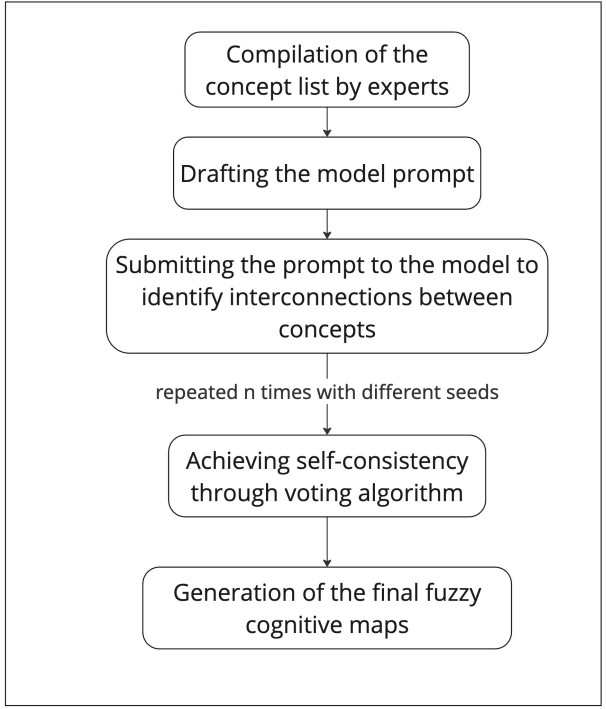

Figure 1: Algorithm for determining concept relationships in FCMs using LLMs.

To identify causal relationships between concepts in the FCMs developed for this study, we systematically tested a range of LLMs, including both proprietary and open-source systems. These included gpt-4.5, O3, Phi-4:14B, Qwen2.5:14B, LLaMA3:8B, Mistral-small, Gemma3:12B, and DeepSeek-R1. Each model was prompted to infer direct causal links between pairs of concepts based on natural language descriptions. Where necessary, concept names were translated into English to ensure prompt clarity and consistency across models.

The motivation for testing multiple models stems from their differing architectures, training regimes, and domain specialisations, which can influence their ability to extract implicit or domain-specific causal knowledge. gpt-4.5 and o3 were selected for their state-of-the-art reasoning capabilities, while Phi-4:14B and DeepSeek-R1 were included for their strong performance in logical inference and long-context understanding. Open-source lightweight models like LLaMA3:8B, Mistral-small, and Gemma3:12B were chosen to support reproducibility and benchmarking. By comparing the outputs of diverse models, we aimed to assess both their individual and relative effectiveness in producing interpretable and accurate FCM structures.

The following query was used to extract causal relationships from text sources:

> Given the following concepts, identify all possible causal relationships between them. For each identified relationship, determine the direction (A → B or B →

---

[1]Algorithm for determining concept relationships in FCMs using LLMs.

A) and assign a weight between -1 and 1, representing the strength and polarity of the relationship. Use Chain-of-Thought reasoning to explain each causal link. You MUST analyse ALL concept pairs. Include medium and low strength relationships in the analysis.

Concepts: concepts

You MUST return ONLY JSON loadable with json.parse(). Provide the output in JSON format with the following structure:

[ "cause": "Caused the influence concept from the list", "effect": "Affected concept from the list", "weight": "between -1 and 1, the strength and polarity of the relationship" ]

Example: [ "cause": "Concept 1", "effect": "Concept 2", "weight": "-0.5" , "cause": "Concept 1", "effect": "Concept 3", "weight": "0.1" ] ...

## 3  RESULTS AND DISCUSSION

In Table[2] we present the performance of eight LLMs across three domains. The evaluation considered four main metrics: number of missing links, correct link predictions, incorrect links, and correct identification of no-link pairs. These metrics provide insight into a model's ability to identify domain-relevant causal relationships (sensitivity), reject spurious associations (specificity), and maintain a balanced trade-off between overprediction and underprediction.

Table 1: Performance of LLMs on FCM link prediction across domains

| FCM name | Total FCM links | Model | Missing links | Correct links | Incorrect links | Correct no link |
|---|---|---|---|---|---|---|
| Urban development | 85 | phi4:14b | 59 | 22 | 8 | **811** |
| | | qwen2.5:14b | 65 | 20 | **5** | 810 |
| | | llama3:8b | 61 | 21 | 17 | 801 |
| | | mistrall-small | 50 | 21 | 42 | 787 |
| | | gemma3:12b | 53 | 21 | 35 | 791 |
| | | deepseek-r1 | 59 | 21 | 21 | 799 |
| | | gpt-4.5 | **43** | 25 | 45 | 787 |
| | | o3 | 47 | **26** | 30 | 797 |
| Retail system | 403 | phi4:14b | 285 | **101** | 32 | 1791 |
| | | qwen2.5:14b | 294 | 95 | **19** | **1801** |
| | | llama3:8b | 288 | 96 | 29 | 1796 |
| | | mistrall-small | 280 | 95 | 43 | 1791 |
| | | gemma3:12b | **276** | **101** | 52 | 1780 |
| | | deepseek-r1 | 290 | 97 | 25 | 1797 |
| | | gpt-4.5 | 281 | 100 | 36 | 1792 |
| | | o3 | 286 | 95 | 34 | 1794 |
| Brain Tumour | 20 | phi4:14b | 9 | 3 | 13 | 56 |
| | | qwen2.5:14b | 13 | 2 | 15 | 51 |
| | | llama3:8b | 18 | 0 | 9 | 54 |
| | | mistrall-small | 7 | **4** | 33 | 37 |
| | | gemma3:12b | 13 | 2 | 14 | 52 |
| | | deepseek-r1 | 19 | 0 | **4** | **58** |
| | | gpt-4.5 | **3** | **4** | 31 | 43 |
| | | o3 | 7 | **4** | 33 | 37 |

The results reveal varied performance patterns across models and domains, offering insight into their reasoning consistency and adaptability to different subject areas.

gpt-4.5 and o3 consistently achieve the highest number of correct links across domains, though they also have a higher false-positive rate. This makes them the most aggressive and recall-focused models within the threshold. They are most useful in applications where capturing all true causal relationships is more important than avoiding false positives.

Gemma3:12b shows a promising balance of recall and precision, especially in the Retail domain. Mistral-small tends to overpredict links in all domains, which may limit its reliability.

---

[2]Performance of LLMs on FCM link prediction across domains

Table[3] shows a noticeable drop in correct link predictions compared to Table[4], which is expected due to the application of a threshold on the connection strength of 0.5. This setting filters out the analysis of low-impact relationships to focus only on the dependencies that significantly affect the system. This is particularly relevant in real-world FCM applications, where weak causal signals can introduce noise and mislead decision-making. By imposing this threshold, we assessed whether LLMs can still accurately recover strong, meaningful causal links across three domains.

Table 2: Performance of LLMs on FCM link prediction across domains with threshold of 0.5

| FCM name | Total FCM links | Model | Missing links | Correct links | Incorrect links |
|---|---|---|---|---|---|
| Urban development | 35 | phi4:14b | 31 | 2 | 2 |
| | | qwen2.5:14b | 35 | 0 | **0** |
| | | llama3:8b | 32 | 0 | 3 |
| | | mistrall-small | 24 | 3 | 8 |
| | | gemma3:12b | 27 | 4 | 4 |
| | | deepseek-r1 | 32 | 1 | 2 |
| | | gpt-4.5 | **17** | **8** | 10 |
| | | o3 | 21 | **8** | 6 |
| Retail system | 73 | phi4:14b | 67 | 4 | 2 |
| | | qwen2.5:14b | 72 | 1 | **0** |
| | | llama3:8b | 72 | 1 | **0** |
| | | mistrall-small | **1** | 4 | 6 |
| | | gemma3:12b | 62 | **7** | 4 |
| | | deepseek-r1 | 71 | 2 | **0** |
| | | gpt-4.5 | 63 | **7** | 3 |
| | | o3 | 69 | 3 | 1 |
| Brain Tumour | 11 | phi4:14b | 5 | 3 | 3 |
| | | qwen2.5:14b | 8 | 2 | 1 |
| | | llama3:8b | 11 | 0 | **0** |
| | | mistrall-small | 4 | **4** | 3 |
| | | gemma3:12b | 7 | 2 | 2 |
| | | deepseek-r1 | 11 | 0 | **0** |
| | | gpt-4.5 | **1** | **4** | 6 |
| | | o3 | 4 | **4** | 3 |

## 3.1 Urban Development

For the FCM of Urban Development, the best-performing models in terms of correct link prediction are gpt-4.5 (25 correct links) and o3 (26 correct links). However, o3 also predicted 30 incorrect links, showing a trade-off between sensitivity and precision. The fewest incorrect links were made by Qwen2.5:14B (5), which still generated 20 correct links, indicating a conservative yet accurate approach. In contrast, mistral-small had the highest number of incorrect links (42), though it correctly predicted 21 links. Phi4:14b showed a balanced performance (22 correct, 8 incorrect) and a high number of correct no-link predictions (811), highlighting its overall cautious and accurate approach to link discrimination.

We selected the gpt-4.5 model for the detailed analysis of the results on the concept level.

Missing dependencies.

These reveal the model's limited sensitivity to key macroeconomic and social dynamics. Examples include:

- Gross regional product $\rightarrow$ unemployment
- Volume of health care financing $\rightarrow$ social tension
- Inflation rate $\rightarrow$ level of demand and consumption

Incorrect links.

These include:

---

[3]Performance of LLMs on FCM link prediction across domains with threshold of 0.5
[4]Performance of LLMs on FCM link prediction across domains

- Inflation rate → economic stability

- Gross regional product → volume of industrial production

- Political stability → volume of investments

These examples suggest the model may overestimate the significance of some factors or oversimplify complex, multifactorial processes.

OVERESTIMATED RELATIONSHIPS.

In some cases, the model correctly identified relationships but overestimated their strength:

- Technological advancement → resource efficiency

- Labour potential → industrial output

- Volume of investments → level of technology development

## 3.2 RETAIL SYSTEM

For the Retail System FCM, the largest with 403 FCM links, phi4:14b and gemma3:12b both achieved the highest correct link count (101), demonstrating strong link prediction capabilities in this more complex setting. However, gemma3:12b also recorded the highest number of incorrect links (52), suggesting it may overpredict links. Qwen2.5:14b showed high precision with only 19 incorrect predictions and a correct no-link count of 1801 which is the highest among all models while having slightly fewer correct links (95). Deepseek-r1 stood out with a solid balance (97 correct, 25 incorrect, 1797 correct no-link), performing consistently well across all dimensions.

Detailed analysis reveals that some causal relationships were correctly identified by the LLM, in agreement with expert assessments. Notable examples include:

- Technological level of equipment → labour productivity

- Lost working hours → labour productivity

- Employee loyalty → staff turnover

- Number of trained personnel → labor productivity

- Development investment → technological level of equipment

OVERESTIMATED LINKS.

- Company reputation → level of employee loyalty

- Rent → all expenses

- Exchange rate → purchase price

This may result from simplified or generalised assumptions that ignore intermediary factors. For instance, although company reputation affects employee loyalty, the actual influence may be more indirect, as loyalty also depends heavily on working conditions, income and openness of communication with employees.

INCORRECT LINKS.

- Foreign investment → development investment

- Accounts payable → working capital

These inaccuracies may arise from underestimating the complexity of economic and organizational processes.

MISSING LINKS.

- Technological level of equipment → number of trained personnel

- Technological level of equipment → product quality

- Speed of technology adoption → product quality

- System integration with suppliers → total expenditure

- Price segment of goods → production standards

- Product quality → company reputation

Omitting these relationships could significantly distort the company's operational and strategic landscape.

## 3.3 BRAIN TUMOUR

The Brain Tumour FCM presents a much smaller and more sensitive map with only 20 links, and it proved to be the most challenging for all models. Most models predicted just a few links correctly, and deepseek-r1 was the only model that made no missing link errors, yet it failed to predict any links correctly. This indicates a highly conservative behaviour—prioritising minimising false positives at the cost of recall. In contrast, gpt-4.5 and o3 each predicted four correct links, although with a high number of incorrect links (31-33), suggesting an aggressive prediction with less precision. Interestingly, gpt-4.5 and phi4:14b both correctly identified 56–58 no-link pairs, with deepseek-r1 achieving the highest score (58), again reinforcing the model's cautious approach. These results highlight a potential limitation in domain adaptation for models not specifically fine-tuned for specialised biomedical contexts, especially when the dataset is sparse. Due to the poor results, a detailed analysis was not conducted.

## 4 CONCLUSION

The analysis LLMs are capable of effectively identifying many economic relationships; however, they still face challenges in capturing complex social and macroeconomic factors. To enhance the accuracy and interpretability of causal modelling, it is important to combine automated analysis with expert evaluation. Future improvements should address current limitations—for example, through the use of pairwise or block-based memory-aware methods to generate more accurate and resilient FCM structures.

Overall, the findings show that model performance varies significantly across domains. Larger models such as phi4:14b and o3 tend to perform better in socio-economic contexts, demonstrating stronger ability to predict dependencies. However, in highly specialised or data-sparse domains like biomedicine, performance tends to decline. Conservative models like Qwen2.5:14B and deepseek-r1 typically demonstrate higher precision by avoiding incorrect links, while more aggressive models like gemma3:12b favour recall, often at the expense of precision.

To improve the quality of FCM construction using LLMs, several factors must be considered. These include the integration of expert validation to assess the significance of predicted links, the development of specialised techniques for adjusting link weights—particularly in management and economic models—and the expansion of model pretraining data with documents describing the relevant domain, especially in the biomedical field. Additionally, deeper analysis of hidden dependencies is needed to minimise omissions of important concepts and connections.

In summary, LLMs show strong potential as tools for automating the analysis of complex systems through FCMs. They can serve as a useful starting point for constructing initial causal structures or act as an additional evaluator to support expert analysis. However, achieving high modelling accuracy and reliability requires their integration with expert knowledge and the adaptation of causal reasoning methods to the specific characteristics of each domain.

## 5  LIMITATIONS

Although the results of this study are encouraging, there are several limitations to consider. The evaluation of FCMs generated by LLM was carried out on a limited set of domains, which may affect the generalisability of the conclusions. The chosen FCMs differ in complexity and subject area, but for a more robust validation of scalability, it would be necessary to include additional fields such as law, education, or environment. Moreover, the use of expert-constructed FCMs as reference introduces a degree of subjectivity, since experts may have different views on causal relationships. While a threshold value (e.g., 0.5) was applied to exclude weaker links, it was chosen empirically and might not be optimal for all domains. The study also assumes that the strength of causal links is accurately reflected by the model's confidence scores, which may not fully correspond to the domain-specific understanding of causality. Furthermore, the generation of FCMs is highly dependent on the specific wording and structure of the prompt text, which can significantly influence the resulting concepts and links. This prompt sensitivity may limit reproducibility and make it challenging to standardise the process across diverse domains. Finally, the models used in the experiments were not specifically fine-tuned for the task of FCM construction, and their performance could potentially be improved through domain-specific training or the development of more structured prompting techniques.

## FUNDING

The research was supported by a grant for the organization of training of students in higher education programs for leading specialists in the field of artificial intelligence, provided by the Analytical Center under the Government of the Russian Federation No. 70-2025-000735 dated May 29, 2025. IGK 000000Ts330325R2Zh0002

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
