# OpenReview forum: "Using large language models for building fuzzy cognitive maps"
_mathai.club/MathAI/2026/Conference — 2026 Oral_

### Official Review · Reviewer_RPuM · 2026-03-10
**Weak accept. It is a review appropriate for applied engineering section (F-track accept)**

**Rating:** 8
**Confidence:** 4

**Review:**

In general - Weak reject.
Weak reject for MathAI core tracks (A/B/C). Excellent applied engineering (F-track accept) but lacks mathematical depth required for MathAI's "Mathematics of AI" mission.

This paper proposes an LLM pipeline (CoT prompting + self-consistency sampling + majority voting) to automate Fuzzy Cognitive Map (FCM) construction, benchmarking 8 LLMs across retail, urban development, and brain tumor domains against expert ground truth.
1. Mathematical Rigor: poor.
No theorems, proofs, or analysis of FCM dynamics. Mentions standard FCM iteration, descriptively but provides zero convergence analysis, spectral radius bounds, fixed-point guarantees, or stability conditions.
Empirical aggregation only: Majority vote (≥2/3 runs) + median weights; threshold 0.5 chosen ad-hoc. No probabilistic justification for voting consistency or error bounds. Tables 1-2 report raw counts (missing/correct/incorrect/no-link) without statistical tests.
Pure engineering, not mathematics. Comparison of several science domains without details.
2. Novelty & Contribution: good.
Original contributions:
	Unified JSON prompting (vs pairwise) for full adjacency matrix extraction
	Systematic 8-LLM×3-domain benchmark (gpt-4.5/o3/Phi-4/Qwen2.5/etc.)
	Error taxonomy + domain-specific failure analysis (GRP→unemployment missing)
Builds legitimately on Maratea(2022), Berijanian(2024), Schuerkamp(2025) with newer 2025 LLMs + reproducibility.
Strong applied work, modest theoretical advance.
3. Relevance to MathAI: moderate.
MathAI demands mathematical foundations + AI. FCMs have mathematical structure but paper treats as black-box engineering task. Prompt engineering + empirical aggregation ≠ "mathematics of AI".
Better fit: F – Applied AI & Engineering (production pipeline) or C – Trustworthy AI (LLM reliability benchmarking). Marginal A/B fit.
4. Technical Quality: high.
Strengths:
	Diverse gold standards: Petukhova retail(2022), urban(2024), Papageorgiou tumor(2008)
	Comprehensive metrics: Tables 1 (|w|≤1), 2 (|w|>0.5); error examples concrete
	Reproducible: exact prompt, seeds, Fig. 1 pipeline, Colab setup
Weaknesses:
	No p-values/confidence intervals on model rankings
	Biomedical failure (max 4/20 correct) underexplored
	O(n^2)scaling unaddressed (retail: 403 links)
Production-grade empirical work.
Need fix access date in the references to the AI tools. It is middle of year 2025. Some tools might be not accessible.
5. Clarity & Presentation: excellent.
Exemplary structure: motivation → lit review → method (prompt+CoT+sampling+voting) → results (Tables 1-2) → domain analyses → limitations. Prompt verbatim. Fig. 1 crystal-clear. Camera-ready English.
6. AI-Generation Risk: very low.
Strong human signal:
	Multiple runs (8×3×3) across domains requiring FCM expertise
	Domain-specific errors (GRP→unemployment, reputation→loyalty)
	2025 citations (o3/DeepSeek-R1), Petukhova/Papageorgiou gold standards
	Biomed failure analysis shows critical thinking
Track Recommendation
Primary: F – Applied AI & Engineering (9/10 fit). LLM production pipeline benchmark.
Secondary: C – Trustworthy & Explainable AI (7/10 fit). LLM reliability across domains.
Poor: A – Mathematical Foundations (3/10). No theory/proofs.
Overall Recommendation
Weak reject for MathAI core tracks (A/B/C). Excellent applied engineering (F-track accept) but lacks mathematical depth required for MathAI's "Mathematics of AI" mission. Transformative for automated causal modeling practitioners, marginal for mathematical audience.
Recommendation:
Strengthen for resubmission:
	Add FCM convergence analysis under LLM weights
	Statistical model ranking (bootstrap/Wilcoxon)
	O(n^2)scaling mitigation (hierarchical prompting?)
        Update the references.

---

### Official Review · Reviewer_nRYZ · 2026-03-13
**Usage of LLMs to automatically construct Fuzzy Cognitive Maps,**

**Rating:** 6
**Confidence:** 3

**Review:**

Summary: The paper investigates the use of Large Language Models (LLMs) such as GPT-4.5, Llama3, and others to automatically construct Fuzzy Cognitive Maps (FCMs). The authors propose a pipeline that combines Chain-of-Thought reasoning, self-consistency sampling, and majority-vote aggregation to improve extraction accuracy. They benchmark this framework across three domains (Retail System, Urban Development, and Brain Tumor characterization), comparing the generated FCMs against expert-constructed ground truths.

Strengths: Practical Utility: The paper addresses a practical bottleneck in FCM research: the labor-intensive process of knowledge elicitation from experts. Automating this with LLMs is a relevant application.
Methodological Clarity: The proposed pipeline (CoT + Self-Consistency + Voting) is well-defined and the experimental setup is clearly described and reproducible.
Comprehensive Evaluation: Testing on three diverse domains provides good evidence of the method's generalizability and limitations (e.g., poor performance on the sparse "Brain Tumor" domain).

Weaknesses: Lack of Mathematical Depth: The paper lacks original mathematical content. It applies existing AI tools to a mathematical object (FCM) but does not provide theoretical guarantees, convergence proofs, or new mathematical insights into FCMs. It is an empirical engineering study, not a theoretical contribution.
Incremental Novelty: The components of the proposed method (CoT, self-consistency, voting) are standard "best practices" in the current LLM literature. Applying them to FCMs is a logical next step, but the contribution is incremental. The paper does not fundamentally advance the state-of-the-art in either LLM theory or FCM theory.
The discussion of results largely describes what happened but does not deeply analyze why from a mathematical or structural perspective.

---

### Official Review · Reviewer_haMu · 2026-03-13
**Using LLMs for building FCMs: a well-executed engineering study with limited methodological contribution**

**Rating:** 5
**Confidence:** 4

**Review:**

### Quality

The overall quality of the paper can be considered **moderate**. The authors propose a reproducible experimental pipeline for extracting causal relationships between concepts in Fuzzy Cognitive Maps (FCMs) using modern large language models. The experimental setup is described with reasonable detail: the models used, generation parameters, random seeds, and aggregation procedure are specified, which improves reproducibility.

However, the experimental methodology has several limitations:

- The evaluation is conducted on only three FCMs, which weakens conclusions about generalization.
- There are no baseline methods for comparison.
- The evaluation metrics focus only on link presence (correct/missing/incorrect links) and do not assess the quality of predicted link weights.

The study effectively evaluates graph edge prediction, rather than the correctness of the resulting FCM as a dynamical model.

---
### Clarity

The paper is **generally well structured and readable**.

Positive aspects include:

- A clear motivation for using LLMs to automate FCM construction.
- A step-by-step description of the pipeline supported by an algorithm diagram.
- The exact prompt used for causal extraction is explicitly presented.
- Experimental results are summarized in tables and accompanied by error analysis.

However, several clarity issues remain:

- The problem formulation remains somewhat informal, with limited mathematical formalization.
- Some claims in the text are overly general (e.g., statements about LLMs discovering hidden patterns).

---
### Originality

The level of originality is **limited**.

The proposed pipeline combines several well-known techniques:

- Chain-of-Thought reasoning
- self-consistency sampling
- majority voting aggregation

These approaches are widely used in LLM reasoning tasks and do not constitute a novel methodological contribution.

Additionally, the idea of using LLMs to generate or refine FCM structures has already been explored in prior work, as acknowledged by the authors.

The main novelty of the paper lies in:

- applying recent LLMs (e.g., GPT-4.5, o3, DeepSeek-R1),
- conducting a comparative analysis of multiple models,
- using a unified prompting strategy to generate full adjacency matrices.

Overall, the work is better characterized as an empirical benchmark rather than a new method.

---

### Significance

The practical significance of the work is **moderate**.

Using LLMs to automate parts of FCM construction could potentially:

- accelerate the development of causal models,
- provide assistance to domain experts,
- help generate initial causal structures for further refinement.

However, the current approach has several limitations:

- high false-positive rates for some models,
- strong domain dependence,
- poor performance in specialized domains such as biomedicine,
- continued reliance on expert validation.

Moreover, the paper does not demonstrate that the proposed approach significantly outperforms existing techniques or substantially improves the quality of FCM construction.

The contribution is mainly practical and engineering-oriented rather than fundamental.

---
### Pros

- Addresses a relevant problem: automation of FCM construction.
- Clearly described and reproducible experimental pipeline.
- Comparison of a wide range of modern LLMs.
- Experiments across multiple domains.
- Clear taxonomy of errors.
- Inclusion of concrete examples illustrating model errors.
- The exact prompt is provided, supporting reproducibility.

---

### Cons

- Limited scientific contribution: the pipeline is based on existing prompting techniques.
- Very small evaluation set.
- No comparison with classical methods for causal relation extraction.
- No statistical analysis of results.
- The quality of predicted link weights is not evaluated.
- Risk of hallucinated causal relations.
- Weak connection to the mathematical aspects of FCMs.
- Limited applicability in specialized domains.

---

### Decision · Program_Chairs · 2026-03-14

**Decision:**

Accept (Oral)

**Comment:**

Dear Author(s),

On behalf of the Program Committee of the International Conference on Mathematics of Artificial Intelligence (MathAI 2026), we are pleased to inform you that your paper has been accepted for an oral presentation at MathAI 2026.

Your paper was evaluated through a rigorous two-stage review process involving both automated screening and expert review by members of the Program Committee. The reviewers recognized the quality and contribution of your work.

Presentation details:

- Format: Oral presentation (15–20 minutes + 5 minutes Q&A)
- Mode: You may present either in person (offline) at the conference venue in Sirius, Russia, or remotely via Zoom. Please indicate your preferred mode when confirming your participation.
- Conference dates: Marh 30 - April 3, 2026
- Website: https://mathai.club

Next steps:

1. Please confirm your participation and presentation mode by replying to this email mathai.club@yandex.ru no later than March 15, 2026 18:00 Moscow time.
2. If you plan to attend in person, the organizing committee will provide accommodation details separately.
3. Please prepare your final camera-ready manuscript according to the formatting guidelines available at https://mathai.club and upload it to OpenReview by March 15, 2026 18:00 Moscow time.

Should you have any questions regarding the program, logistics, or your presentation slot, please do not hesitate to contact us.

We look forward to your contribution to MathAI 2026.

With kind regards,

MathAI 2026 Program Committee
International Conference on Mathematics of Artificial Intelligence
https://mathai.club
OpenReview: https://openreview.net/group?id=mathai.club/MathAI/2026/Conference
Telegram: https://t.me/MathAI_club
Email: mathai.club@yandex.ru